# The Correlations among Dietary Lifestyle, Microecology, and Mind-Altering Toxoplasmosis on the Health of People, Place, and the Planet

**Vanessa de Araujo Goes** [1,2,*] **, Yusuf Amuda Tajudeen** [2,3,4] **and Mona Said El-Sherbini** [2,5,*]

1   LeBioME-Bioactives, Mitochondria and Placental Metabolism Core, Institute of Nutrition Josué de Castro, Federal University of Rio de Janeiro, Rio de Janeiro 21941-902, Brazil
2   Member of the Nova Network, Baltimore, MD 21231, USA
3   Department of Microbiology, Faculty of Life Sciences, University of Ilorin, PG. 1515, Ilorin 240003, Nigeria
4   Department of Epidemiology and Medical Statistics, Faculty of Public Health, College of Medicine, University of Ibadan, P.M.B 5017 G.P.O, Ibadan 200212, Nigeria
5   Department of Medical Parasitology, Faculty of Medicine, Cairo University, Cairo 11562, Egypt
*   Correspondence: vgnutricao@gmail.com (V.d.A.G.); monas.elsherbini@kasralainy.edu.eg (M.S.E.-S.)

**Abstract:** Being one of the most common foodborne protozoa worldwide, chronic toxoplasmosis caused by *Toxoplasma gondii* (*T. gondii*) could contribute significantly to the etiology of several mental disorders. The neurotropic parasite can directly influence the gut microbiota, causing inflammation with subsequent degradation of tryptophan required for parasite growth. Research in humans and animals shows that the gut microbiome is involved in the regulation of brain serotonergic pathways through the microbiota–gut–brain axis. Since the serotonin system is extensively interconnected with the body's master clock through neuronal networks, the microbiota has been suggested as a potential mediator, fine-tuning circadian misalignment, following a reciprocal relationship with human eating patterns. Furthermore, adherence to an intermittent fasting diet can improve the serotonin biosynthesis pathway in the intestines and improve cognitive function. This review aims to explain the role of fasting in parasite-driven gut microbiome perturbation and the mechanisms by which *Toxoplasma* infection alters brain function. Due to its significant impact on social–economic status, diet patterns, microbiota disruption, circadian rhythm, chronic inflammation, and mental disorders, toxoplasmosis is an underestimated threat that could be prevented by simple lifestyle changes through educational actions. Furthermore, there are few research studies that address toxoplasmosis-induced mental disorders from a holistic perspective. Thus, a planetary health lens is needed to understand these correlations that directly relate to the promotion of a resilient and empathic civilization, crucial to enabling a flourishing healthy society on all scales.

**Keywords:** dietary patterns; lifestyle; toxoplasmosis; microbiome; mental health; planetary health

## 1. Introduction

Being one of the most common foodborne parasitic infections worldwide, toxoplasmosis caused by *Toxoplasma gondii* represents a significant and emerging global public health burden [1]. Globally, an estimated 750 deaths are attributed to toxoplasmosis per year, and 50% of these deaths are related to eating meat contaminated with tissue cysts—the infective stage of toxoplasmosis [2,3]. The pathology and epidemiology of chronic toxoplasmosis emphasizes the relevant aspects of individual, place, and planet health and their correlation with anthropogenic driving factors. These macroecological factors contribute to the widespread transmission of toxoplasmosis infections with negative consequences on global mental health, as will be discussed in this article.

The obligatory intracellular tissue protozoan *T. gondii* has different routes of transmission during its life cycle since it is a heteroxenous and polyxenous parasite. Infective oocyst

stages can be found in contaminated raw vegetables and unpasteurized milk, while viable tissue cyst stages can be found in undercooked meat, all of which can cause oral infection with toxoplasmosis. This directly renders some traditions around cultural eating habits a risk factor related to toxoplasmosis. Additionally, the ability of *Toxoplasma* tachyzoites stages to cross the placental barrier in IgM-positive pregnant women further exacerbates health risks in future generations through vertical transmission (transgenerational influence). Due to the different routes of transmission of toxoplasmosis—oral, hematogenous, transplacental—it is estimated to infect up to a third of the global population [1,4].

The onset of toxoplasmosis infection is often asymptomatic and therefore inadequately managed in most cases. However, serious health problems represent an underestimated threat, especially in neonates and immunosuppressed patients [4]. Once chronic/latent toxoplasmosis is established as indicated by seropositivity of anti-toxoplasma IgG antibodies, the tissue cyst stages detected in the brain and skeletal muscles can persist for life in humans, who act as an intermediate host in the parasite cycle. The first report on the correlation between latent toxoplasmosis and human cognitive function alteration was in 1953 [5]; in the following decades, more evidence was gathered through controlled behavioral analysis in humans and experimental animal models that reported mental alterations and host behavior change, psychiatric disorders, and neuropathies, to mention just a few [6–8]. The mechanisms of neuropathology vary in chronic toxoplasmosis of the brain, which has been reported to be due to brain inflammation, hormonal and neurotransmitter changes, or both [9]. The implications of mental health disorders caused by toxoplasmosis in our society and global healthcare systems can be significantly underestimated given their widespread prevalence worldwide.

Toxoplasmosis has a complex pathogenesis; following a foodborne infection, at the tachyzoite stage, it replicates in the enterocytes of the human gut. This stimulates the production of pro-inflammatory cytokines, chemokines, and immune mediators that disrupt the immune system [10]. This microecological inflammatory status leads to bacterial dysbiosis, immune-mediated tissue damage, and inflammation in the intestinal tract [11]. Furthermore, the *Toxoplasma* parasite itself alters different neurotransmitter metabolism and signaling such as dopamine, glutamate, γ-aminobutyric acid (GABA), and serotonin [12]. The intracellular nature of *Toxoplasma* tachyzoites stimulates different immune responses, yet the interactions with the gut microbiota need to be further elaborated [11,12]. Additionally, the gut–brain axis (GBA), a critical component of brain function and human behavior, is influenced by microbiome composition and diversity, a fact which has made substantial contributions to our understanding of the physical and mental health in a more integrated approach [11,13]. Hence, through this biochemical and physiological integration, the microbiome can be a key player in mitigating *Toxoplasma* infection, especially in the context of mental health.

Moreover, chronic brain infection with tissue cyst formation sustains the inflammatory response, leading to serotonin deficiency by shifting tryptophan metabolism. Tryptophan is an essential amino acid used for protein synthesis and is the sole precursor for the neuroendocrine transmitter serotonin and the pineal hormone melatonin [14]. There is evidence that changes in the composition of the gut microbiota affect GBA by modulating tryptophan metabolism [14]. Furthermore, products of tryptophan metabolism, such as serotonin, kynurenine, and indole compounds, are signaling molecules between gut microbiota and brain functions [15].

The gut microbiome plays a critical role in regulating host immunity not only through its key metabolic outputs, such as short chain fatty acids and tryptophan metabolites, but also via its interaction with the circadian rhythm [16,17], which is a central immune regulator. The bacterial community oscillates in composition, biophysical location, and function over a period of 24 h, driving rhythmic metabolic processes [18]. This intrinsic circadian machinery in the gut microbiota has many implications in host–microbiome interactions and, hence, on the entire function of the immune system.

Within the drivers of the microbiome, circadian variation (rhythmicity) is the timing of meals and the dietary composition, which is controlled by the host and influenced by the availability of food and social and cultural norms [19]. Microbiota itself can modulate the master clockwork machinery through its metabolic outputs.

There is evidence that the intermittent fasting regimen (IF) is associated with health benefits through different mechanisms [20]. The fasting/feeding dietary rhythm is suggested to stimulate the fluctuation of our gut microbiota and, therefore, a series of molecular alterations which in turn restore our inherent healthy circadian clock that has been disrupted by our superimposed anthropogenic lifestyle [21].

This article reviews the correlations among diet patterns, specifically intermittent fasting, the gut microbiome, circadian rhythm, and mental health in the context of toxoplasmosis, with an emphasis on the correlation between dietary lifestyle and foodborne illness. A holistic viewpoint from a micro- to a macro-ecological approach is highlighted in the following sections to understand the impact of toxoplasmosis on an individual, societal, and global level.

## 2. Lifestyle and Anthropogenic Drivers on *Toxoplasma*–Microbiome Microecology

Today, in the Anthropocene context, it is imperative to highlight the interdependency among lifestyle, ecosystems, and infectious disease. Despite ongoing infectious disease control programs, toxoplasmosis still accounts for 3 of the top 10 causes of death worldwide in the 21st century, especially in immunosuppressed populations [22]. Industrialization and modern society brought about a paradigm shift in lifestyle that led to misalignment in various pillars such as diet, exercise, sleep, mental well-being, and environmental exposure that has been associated with the burden of many diseases, including infectious ones [23]. Human activities, behaviors, and culture, including eating patterns, interfere with an individual's ecosystem down to its microbial foundation [24]. At the same time, land use, food production, global trade, and travel are considered anthropogenic drivers of disease emergence, several of which are shared drivers of biodiversity loss, environmental degradation, and ecosystem disruption [23]. These scenarios facilitate the wide environmental spread of toxoplasmosis in different populations and age groups [24,25]. Growing evidence supports the potential role of the exposome concept (lifestyle and environmental exposures since birth) in the prevention or improvement of disease susceptibility [26].

On a microscale, the microbiota, characterized as a diverse set of microbes that colonize the human gut, represents the first line of resistance against any infection, including parasitic ones, modulating the susceptibility to and severity of infections through different mechanisms [27]. Protection against pathogen colonization is through niche exclusion and bacterial products that influence pathogen infectivity, its nourishment, and the release of microbiome-derived substances, all of which subsequently stimulate innate and adaptive immune responses [28]. Furthermore, some of these microbiota-derived metabolites curb infection through tryptophan breakdown products (indole and its derivatives) that modulate the mucosal barrier and enhance the host immune response. Thus, these neurochemical metabolites can inhibit pathogen growth and/or virulence pathways [27].

The development of infection signifies the crucial role of the microbiome in regulating the host physiology, as described by many scholars [17,27,29]. However, the combined internal infections and external factors such as lifestyle and anthropogenic exposomes that disrupt the microbiota's function and diversity (dysbiosis) further interfere with the intrinsic immunological cascade, resulting in chronic inflammation and fulminant pathology depending on the immune system's overall health [30].

At the epidemiological level, *T. gondii* tachyzoites are capable of infecting any nucleated cell, and almost all warm-blooded vertebrates can act as intermediate host for the parasite. On a demographic scale, its prevalence is dictated by social–economic status (hygiene, sanitation) and diet patterns. However, the interplay of the microbiota and circadian rhythm disruption, as well as chronic inflammation, all contribute to immune health and can be captured within the biopsychosocial aspect of the individual [31]. The modern

lifestyle has influenced the structure–function relationship between the microbiota and infectious diseases in various ways [27]. For example, disruption of circadian rhythm is echoed in gut microbiome dysbiosis; therefore, this interdependency, which is influenced by many components of the exposome (sleep, diet), has an important, underestimated role in susceptibility to toxoplasmosis [32].

Additionally, since the microbiota modulates brain function by influencing serotonin metabolism, its disruption adds to the parasite tropism for the brain, which may increase mental disorders [15]. Hence, characterizing toxoplasmosis as an important disease within a holistic-systems-based perspective of planetary health can be insightful. Planetary health considers all natural and anthropogenic drivers of human health, physical and mental alike, calling for the promotion of a resilient and empathic civilization, which is crucial to enable a flourishing society [25].

### 3. Biochemical Correlations between Toxoplasmosis and Mental Health

Associations between toxoplasmosis and various neuropathies and psychiatric disorders have been reported in IgG seropositive patients with chronic toxoplasmosis. This includes behavioral changes and depression characterized by lower levels of serotonin [22]. Tryptophan, the precursor of serotonin, is essential for *T. gondii* growth. Biochemically, inflammation caused by *T. gondii* upregulates indoleamine-2,3-dioxygenase (IDO) and tryptophan-2,3-dioxygenase (TDO), leading to a catabolic tryptophan shunt related to depression. Furthermore, *T. gondii* infection activates the kynurenine pathway, resulting in high levels of neurotoxic metabolites such as kynurenic acid (KYNA) and quinolinic acid (QUIN). High levels of KYNA influence dopamine and glutamate levels, altering cognitive functions [22]. Recent studies suggest that toxoplasmosis disrupts the activity of the hypothalamic–pituitary–adrenal gland axis [9] and hormonal disorders, including serotonin, which could lead to behavioral disorders [22,33].

The gut–brain axis (GBA) is a bidirectional communication network between the gastrointestinal tract (enteric nervous system) and the central nervous system. Studies in animal models indicate that the gut microbiota can metabolize tryptophan, affecting its availability to the host, as well as its products, such as serotonin, kynurenine, tryptamine, and indole compounds, which have effects on this complex crosstalk [31,34]. Changes in the gut microbiota composition affect the GBA through modulation of tryptophan metabolism, including serotonin synthesis and the degradation pathway of the host, such as kynurenine with consequences for brain function [35]. Shao et al. [36] showed that *T. gondii* infection alters gut microbes in mice, suggesting that the gut microbiome alters disease progression. Manipulation of the gut microbiota could be a good strategy to balance tryptophan metabolism and its availability to the host (Supplementary Materials File).

### 4. Dietary Patterns as a Tool to Mitigate Toxoplasmosis

Circadian rhythm or chronobiology is a field of science that integrates the 24 h physiological changes to natural day and night cycles governed by planet Earth's rotation [16]. Although the circadian pattern in human immune cells and hormonal regulation is basically cell-autonomous, infection and environmental factors such as light, temperature, exercise, as well as the dietary fasting/feeding cycle, significantly affect it [20]. Concurrently, the intrinsic circadian master clock of the host is correlated with the peripheral rhythmicity of the gut microbiota [20]. Similarly, the gut microbiota is characterized by interindividual variability due to genetic and environmental factors [35]. Diet is one of the individual extrinsic factors that has a strong influence on gut microbiota composition [37], as well as on the overall regulation of circadian rhythm, mainly through food composition and timing [38]. These delicate interconnections, namely diet, circadian rhythm, and gut microbiota [39,40], are affected by significant changes due to an anthropogenic lifestyle [38]. Behavioral dietary patterns have shifted towards a culture of overconsumption, especially of processed meat, ultraprocessed products, and refined carbohydrates, together with low consumption of vegetables, fruits, grains, and natural whole foods rich in fibers, micronutri-

ents, and bioactive compounds, all of which is grossly reflected in the health and diversity of the gut microbiome to a large extent [37].

A dietary regime involving intermittent fasting is characterized by periods of food abstinence that are longer than the overnight fast [38]. It can be divided into three different patterns: alternate day fasting, whole day fasting, and time-restricted feeding [38]. Food timing modulation through intermittent fasting has been shown to affect the gut microbial population [41,42] through different mechanisms, such as increasing microbiome β-diversity and changing its microbial composition as well as the levels of its metabolites [43,44]. Increased plasma levels of serotonin, tryptophan, bile acids, and short-chain fatty acids (acetate butyrate and propionate) are among the metabolites modulated by the abundance of bacteria in the gut after intermittent fasting [44].

Since circadian rhythmicity modulates every aspect of any living organism, including bacteria from the intestinal microbiome, as well as the single-celled protozoa that inhabit host brain tissues, such as *T. gondii* [45], a time-restricted feeding regimen can restore the diurnal rhythmicity and compositional and functional structures of the intestinal microbiome over the course of the day [38,46,47].

Another aspect of intermittent fasting practices that is related to positive results is calorie restriction, which has been reported to promote various health benefits, including modulation of the immune response through activation of autophagy that improves immunity to fight infection [48]. Furthermore, fasting/feeding cycles activate different metabolic pathways that regulate rhythmic transcriptomes and thus translate into healthier phenotypes [20]. Being an important modulator of circadian rhythm, metabolic pathways, as well as the immune system, diet could be an interesting strategy to mitigate toxoplasmosis disease outcomes (Supplementary Materials File). The impact of intermittent fasting on alimentation is obvious, yet continued research on the therapeutic benefit of such an intervention in reducing the overall psychological morbidity due to toxoplasmosis is still needed to elucidate such correlations in human studies.

## 5. Conclusions

The neurotropic predilection of *Toxoplasma gondii* has received little attention in the global context of mental health, although with a worldwide prevalence. Mental disorders induced by latent toxoplasmosis are caused by the presence of *T. gondii* tissue cysts in the brain, which can persist for years. Insights into the foodborne nature of toxoplasmosis and the mechanisms of parasite manipulation through dietary patterns while integrating the biochemical and microbiota role are worth further investigations. It is imperative to adopt a holistic approach to understand toxoplasmosis in its microecology and pathophysiology. Adopting a holistic approach such as planetary health to address the interconnected challenges of toxoplasmosis linked to lifestyle risk factors will allow for appropriate design of baseline data of the occurrence of the disease at the interface between humans, animals, and the environment—which will allow for adequate and effective decisions for disease control and management.

The widespread nature of toxoplasmosis calls for addressing culture and lifestyle, education, and preventive measures as powerful tools to overcome this mental health threat. Strategies such as examining the relational values of the micro- and macroenvironment and promoting healthy dietary habits and patterns, along with good hygienic practices coupled with robust surveillance studies and translational research on community-wide level, could significantly reduce the global burden of toxoplasmosis and promote the overall health of individuals and societies.

**Supplementary Materials:** The following supporting information can be downloaded at: https://www.mdpi.com/article/10.3390/challe13020063/s1, Figure S1: Gut-Microbiome-Brain-Circadian rhythm interconnections. The neuro-immuno-endocrine adaptative systems. The entry of the parasite into the intestinal tract leads to dysbiosis and inflammation [11], disrupting the immune system as well as brain function through the gut-brain axis (GBA). Toxoplasma infection leads to serotonin and melatonin biosynthesis disruption through a tryptophan catabolic shunt. Furthermore, the gut

microbiota have an intrinsic circadian machinery which rhythmicity is driven by, among other things, the timing of meals and the dietary composition Microbiota itself can modulate the master clockwork, interacting with the neuro-immune-endocrine systems.

**Author Contributions:** Conceptualization, V.d.A.G. and M.S.E.-S.; methodology, V.d.A.G., Y.A.T. and M.S.E.-S.; data curation, V.d.A.G., Y.A.T. and M.S.E.-S.; writing—original draft preparation, V.d.A.G., Y.A.T. and M.S.E.-S.; writing—review and editing, supervision, M.S.E.-S. All authors have read and agreed to the published version of the manuscript.

**Funding:** This research received no external funding.

**Data Availability Statement:** Not applicable.

**Acknowledgments:** The abstract of this article was presented at the 2021 International inVIVO Planetary Health Conference. We are grateful to the organizers and audience for their suggestions and comments, which have helped in improving the quality of the manuscript. We also acknowledge and thank the reviewers and editors of the *Challenges* journal for their effort in enriching this manuscript.

**Conflicts of Interest:** The authors declare no conflict of interest.

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
