# Peer review of "The Correlations among Dietary Lifestyle, Microecology, and Mind-Altering Toxoplasmosis on the Health of People, Place, and the Planet"

_challenges, doi:10.3390/challe13020063_

Round 1
Reviewer 1 Report
revisar a literatura
Author Response
Comments and Suggestions for Authors.
Dear Reviewer,
Thank you for taking the time to review our manuscript, your insightful suggestions and comments have, indeed, strengthen our manuscript.
Thank you.
Sincerely,
Authors.
Reviewer 2 Report
Title
I would suggest a slight modification of the title to “The correlations among Dietary Lifestyle, Microecology, and Mind-altering Toxoplasmosis on the Health of People, Place and the Planet
Abstract
Line 16-37- The abstract is too long. Reduce to at most 250 words. You can do so by reducing the background which is to lengthy.
1 Introduction
Line 41- Reference 1 should be at the end of the sentence.
Line 47- change interconnection to “correlation” and change in any other places
Line 51-54: This sentence is not clear. Rephrase
Line 83- mitigating or metigating?
2 Lifestyle and anthropogenic drivers on toxoplasma-microbiome microecology
Line 115- use “among” instead of “between”
Line 116-118- Rephrase this sentence to bring out the meaning
Line 122-125- inset a reference at the end of this sentence.
Line 161- Biochemical connections between toxoplasmosis and mental health” I would suggest you modify this to Biochemical correlations between toxoplasmosis and mental health
Line 180- put a comma after host
Line 181- “Shao et al.,” not “Shao et al,”
4. Dietary patterns as a tool to mitigate toxoplasmosis.
Line 202- insert a reference
Line 212-214- This sentence is not complete or not clear.
Line 225 Modify “5. Conclusion and prospects - Promotion of the PH approach (education/ life rhythms) 225” to just “Conclusion”
Line 233- The figure should be moved inward into the body of the work, not at the conclusion. This figure 1 was not also shown.
Line 235-238- This sentence is not clear
Line 238- modify “Adopting an holistic approach” to “Adopting a holistic approach”
Author Response
Dear Reviewer,
Thanks so much for reviewing our manuscript and for providing insightful suggestions to improve its quality, we sincerely appreciate this. Here below is how we have revised the manuscript in line with your suggestion and we hope that you are satisfied with our final revision.
Comments and Suggestions for Authors
Title
I would suggest a slight modification of the title to “The correlations among Dietary Lifestyle, Microecology, and Mind-altering Toxoplasmosis on the Health of People, Place and the Planet
Response: This has been addressed in the manuscript and thank you for your suggestion.
Abstract
Comment: Line 16-37- The abstract is too long. Reduce to at most 250 words. You can do so by reducing the background which is to lengthy.
Response: The abstract has been reduced to fit the word count of the journal and has been reduced to 233 words. The changes made can be seen from line 16-34.
1 Introduction
Comment: Line 41- Reference 1 should be at the end of the sentence.
Response: This has been addressed in the manuscript.
Comment: Line 47- change interconnection to “correlation” and change in any other places
Response: This has been addressed in line 43 of the revised manuscript as well as other places as suggested.
Comment: Line 51-54: This sentence is not clear. Rephrase
Response: This has been addressed in the revised manuscript and this can be seen in line 47-50.
Comment: Line 83: Mitigating or metigating?
Response: The manuscript has been revised and this has been addressed in line 86.
- Lifestyle and anthropogenic drivers on toxoplasma-microbiome microecology
Comment: Line 115- use “among” instead of “between”
Response: The manuscript has been revised in line with your suggestion in Line 120.
Comment: Line 116-118- Rephrase this sentence to bring out the meaning
Response: This has been addressed in line 120-122
Comment: Line 122-125- inset a reference at the end of this sentence.
Response: This has been addressed in lines 128-130 as new reference has been added.
Comment: Line 161- Biochemical connections between toxoplasmosis and mental health” I would suggest you modify this to Biochemical correlations between toxoplasmosis and mental health
Response: In line 170, “connections” has been changed to correlations.
Comment: Line 180- put a comma after host
Response: This has been addressed in line 189, thank you.
Comment: Line 181- “Shao et al.,” not “Shao et al,”
Response: This has been revised and changes have been made in line 190.
- Dietary patterns as a tool to mitigate toxoplasmosis.
Comment: Line 202- insert a reference
Response: reference has been inserted in line 202—which is now line 211.
Comment: Line 212-214- This sentence is not complete or not clear.
Response: This has been revised and can be seen in Line 221-223.
Comment: Line 225 Modify “5. Conclusion and prospects - Promotion of the PH approach (education/ life rhythms) 225” to just “Conclusion”
Response: This has been addressed in the line 237.
Comment: Line 233- The figure should be moved inward into the body of the work, not at the conclusion. This figure 1 was not also shown.
Response: The figure is now mentioned in the body of the manuscript in line 193 and has been attached in a supplementary file.
Comment: Line 235-238- This sentence is not clear
Response: The manuscript has been revised.
Comment: Line 238- modify “Adopting an holistic approach” to “Adopting a holistic approach”
Response: This has been addressed in line 245.
We hope that you are satisfied with our revision.
Sincerely,
Authors
Reviewer 3 Report
This is a novel study.
Author Response
Comments and Suggestions for Authors.
Dear Reviewer,
Thank you for taking the time to review our manuscript, your insightful suggestions and comments have, indeed, strengthen our manuscript. We also appreciate your kind words towards our manuscript.
Thank you.
Sincerely,
Authors.
Reviewer 4 Report
This is an interesting topic with some well written sections.
It would be interesting to learn more about the shortcoming/side effects/limitations of intermittent fasting as the review seems to present only the positive aspects of such approach.
A large part of the articles cited are reviews themselves. The article would have a stronger scientific impact if it were to discuss the data obtained from original research papers instead.
Make sure references are suitable, e.g. ref 1 does not support the the fact that toxoplasmosis is one of the most common parasitic zoonosis worldwide and even if it did, it would be based on data that are over 20 years old. Same as ref 3 (Toxoplasma gondii in the Food Supply), which does not support the global estimation of deaths.
Avoid long sentences e.g line 62 to 65, 141-145, 146-50, 226-29
The link between circadian rhythms and toxoplasmosis prevalence needs to be supported by data
Provide references and details for facts reported. e.g. line 68, what is the worldwide prevalence of mental health disorders due to toxoplasmosis? The only data on toxoplasmosis reported is on yearly deaths and it is rather low.
Some sentences don't make sense: e.g. line 71-72, 82-84, 140-141,229-30
Line 155 156, please explain how.
ref needed at the end of several sentences, for example: line 75-76 and inflammation in the intestinal tract.REF and line 77-78 GABA) and serotonin (ref).
The article needs to avoid repeating general statements and give more precise and supported analysis of facts and their meaning. It should also identify limitations in the interpretations of the data. Line 230-41 are very vague.
What is transdisciplinary epidemiology?
Very little of the conclusion and prospects is specific to toxoplasmosis
Author Response
Dear Reviewer,
Thanks so much for reviewing our manuscript and for providing insightful suggestion to improve its quality, we sincerely appreciate this. Here below is how we have revised the manuscript in line with your suggestions and we hope that you are satisfied with our final revision.
Comments and Suggestions for Authors
This is an interesting topic with some well written sections.
It would be interesting to learn more about the shortcoming/side effects/limitations of intermittent fasting as the review seems to present only the positive aspects of such approach.
A large part of the articles cited are reviews themselves. The article would have a stronger scientific impact if it were to discuss the data obtained from original research papers instead.
Make sure references are suitable, e.g. ref 1 does not support the fact that toxoplasmosis is one of the most common parasitic zoonosis worldwide and even if it did, it would be based on data that are over 20 years old. Same as ref 3 (Toxoplasma gondii in the Food Supply), which does not support the global estimation of deaths.
Avoid long sentences e.g line 62 to 65, 141-145, 146-50, 226-29
Response:
The manuscript has been revised in line with your suggestions and highlighted changes have been tracked. Redundant sentences have been removed; new references have been added to make the article suitable, and the limitation of fasting has been included.
Comment:
The link between circadian rhythms and toxoplasmosis prevalence needs to be supported by data
Response:
The manuscript has been revised and the necessary studies have been added and changes made are tracked.
Comment:
Provide references and details for facts reported. e.g. line 68, what is the worldwide prevalence of mental health disorders due to toxoplasmosis? The only data on toxoplasmosis reported is on yearly deaths and it is rather low.
Response: this has been addressed in line 69-72
Comment: Some sentences don't make sense: e.g. line 71-72, 82-84, 140-141,229-30
Response: The manuscript has been revised scientifically and all sentences have been reworded alongside evidence-based studies to augment the points.
Comment: Line 155 156, please explain how.
Response: this has been addressed in line 163-164 and to augment our point here, we have cited a paper.
Comment: ref needed at the end of several sentences, for example: line 75-76 and inflammation in the intestinal tract. REF and line 77-78 GABA) and serotonin (ref).
Response: This has been addressed.
Comment: The article needs to avoid repeating general statements and give more precise and supported analysis of facts and their meaning. It should also identify limitations in the interpretations of the data. Line 230-41 are very vague.
Response:
The manuscript has been revised and necessary changes have been made.
Comment:
What is transdisciplinary epidemiology?
Very little of the conclusion and prospects is specific to toxoplasmosis
Response: The manuscript has been thoroughly revised and necessary changes have tracked.
We hope that you are satisfied with our revision.
Sincerely,
Authors